# Removal of Emerging Organic Pollutants by Zeolite Mineral (Clinoptilolite) Composite Photocatalysts in Drinking Water and Watershed Water

Pengfei Zhou [1,2,3], Fei Wang [1,2], Yanbai Shen [3], Xinhui Duan [1,2], Sikai Zhao [3], Xiangxiang Chen [4] and Jinsheng Liang [1,2,*]

1   Key Laboratory of Special Functional Materials for Ecological Environment and Information, Hebei University of Technology, Ministry of Education, Tianjin 300130, China; pengfeiz0305@163.com (P.Z.); wangfei@hebut.edu.cn (F.W.); dxh1191984@aliyun.com (X.D.)
2   Institute of Power Source and Ecomaterials Science, Hebei University of Technology, Tianjin 300130, China
3   School of Resources and Civil Engineering, Northeastern University, Shenyang 110819, China; shenyanbai@mail.neu.edu.cn (Y.S.); zhaosikai@mail.neu.edu.cn (S.Z.)
4   Zijin School of Geology and Mining, Fuzhou University, Fuzhou 350108, China; chen@fzu.edu.cn
*   Correspondence: liangjinsheng@hebut.edu.cn; Tel.: +86-22-2658-2575

**Abstract:** One of the most challenging problems for people around the world is the lack of clean water. In the past few decades, the massive discharge of emerging organic pollutants (EOPs) into natural water bodies has exacerbated this crisis. Considerable research efforts have been devoted to removing these EOPs due to their biotoxicity at low concentrations. Heterogeneous photocatalysis via coupling clay minerals with nanostructured semiconductors has proven to be an economical, efficient, and environmentally friendly technology for the elimination of EOPs in drinking water and watershed water. Natural zeolite minerals (especially clinoptilolites) are regarded as appropriate supports for semiconductor-based photocatalysts due to their characteristics of having a low cost, environmental friendliness, easy availability, co-catalysis, etc. This review summarizes the latest research on clinoptilolites used as supports to prepare binary and ternary metal oxide or sulfide semiconductor-based hybrid photocatalysts. Various preparation methods of the composite photo-catalysts and their degradation efficiencies for the target contaminants are introduced. It is found that the good catalytic activity of the composite photocatalyst could be attributed to the synergistic effect of combining the clinoptilolite adsorbent with the semiconductor catalyst in the heterogeneous system, which could endow the composites with an excellent adsorption capacity and produce more $e^-/h^+$ pairs under suitable light irradiation. Finally, we highlight the serious threat of EOPs to the ecological environment and propose the current challenges and limitations, before putting the zeolite mineral composite photocatalysts into practice. The present work would provide a theoretical basis and scientific support for the application of zeolite-based photocatalysts for degrading EOPs.

**Keywords:** zeolite mineral (cliloptilolite); composite photocatalysts; semiconductor; emerging organic pollutants

## 1. Introduction

Booming global industrialization and urbanization has caused a high amount of pollution in water bodies worldwide [1–3]. The large discharge of substandard wastewater would pose a serious threat to human beings and other living species [4,5]. People are facing urgent demands to remove pollutants from municipal, agricultural, and industrial wastewater in order to produce domestic water [6]. Generally, common heavy metals ($Pb^{2+}$, $Cu^{2+}$, $Cd^{2+}$, $Ni^{2+}$, and $Mn^{2+}$) [7,8], ammonium ($NH_4^+$), and other ions [9] remaining in wastewater can be easily removed by adsorption. However, emerging organic pollutants (EOPs), including pharmaceuticals and personal care products (PPCPs) [10], persistent

organic pollutants (POPs) [11], disinfectant by-products (DBPs) [12], endocrine disruptors compounds (EDCs) [13,14], antibiotics [15], and microplastics (MPs) [16,17], cannot be decomposed after adsorption, which would lead to the long-term pollution of the environment. Particularly, agricultural runoff, domestic waste, sewage overflow, and the partly metabolized pharmaceuticals excreted by humans and animals bring high EOP emissions to an urban environment [18]. Although the wastewater might meet the discharge standard through wastewater treatment plants (WWTPs), persistent pollutant residues are still detected at the ng/L or mg/L level in surface water and groundwater. To achieve deep water purification, considerable research efforts have been devoted to exploring more highly effective techniques for the removal of EOPs [19]. In the past few decades, conventional physical [20], chemical [21], and biological [22] techniques have been applied for eliminating pollutants. However, some kinds of contaminants with complex molecular structures such as antibiotics and pesticides could not be effectively eliminated in a short time due to certain inherent limitations in these methods. Compared with traditional treatment methods, advanced oxidation processes (AOPs) have become a research hotspot because of their efficient generation of active species to treat complex pollutants [23,24].

Photocatalytic technology, a kind of representative AOP, is known as one of the most promising methods for practical application in removing hazardous pollutants [25]. As a cost-effective technique, photocatalysis can promote the generation of reactive oxygen species (ROS) in the catalyst system to accelerate the photodegradation process [26]. Especially, semiconductor-based photocatalysis is widely applied in research toward contaminant removal [27]. Numerous works in the literature have reported that $TiO_2$ [28], ZnO [29], $SnO_2$ [30], $g$-$C_3N_4$ [31], $MoS_2$ [32], and Bi-based semiconductors [33,34] could be used as photocatalysts to decompose organic pollutants. Moreover, some kinds of metal–oxide–semiconductor (MOS) and their related sulfide, such as CuO/CuS [35,36], NiO/NiS [37,38], $SnO_2$/$SnS_2$ [39,40], etc., can also play the role of photocatalyst in degrading organics under suitable light irradiation. However, pristine MOS photocatalysts such as $TiO_2$ and ZnO have the inherent drawbacks of fast electron/hole pairs recombination and can only be motivated by UV light irradiation owing to their wide bandgap energies. [41,42]. The metal ion doping of a pure MOS photocatalyst is recognized as one of the most effective approaches to shortening the bandgap by inducing the dopant impurity level, and, thus, can enlarge the spectral response range. Moreover, metal dopants can act as electron trap to prevent electron/hole recombination [43]. Another common issue is the severe agglomeration of nano-scale semiconductor catalysts in the process of synthesis and application, which makes recycling extremely difficult after being added to the wastewater [44].

In recent years, heterogeneous photocatalysis has attracted considerable attention as an alternative and environmentally friendly process for the degradation of various pollutants [45]. In theory, the issues of agglomeration and recycling for the pristine semiconductor catalysts can be solved by using substrate materials. To be utilized as support for semiconductor catalysts, the alternative materials should have a strong binding force with semiconductor nanoparticles (NPs) and meet basic requirements such as having a good chemical stability, and excellent adsorption capacity and mechanical strength. Natural clay minerals are the desired candidate carriers because they have abundant adsorption and reaction activation sites and a stable crystal structure, which are very conducive to catalytic performance [46–48]. Among the various clay minerals, zeolite is a popular candidate support owing to its low cost, wide distribution, high adsorption capacity, and high surface area [49,50]. Its unique advantages are mainly reflected in four aspects, as follows: (1) the adsorption ability of composite photocatalysts can be enhanced significantly owing to the existence of abundant micropores, which can enrich the pollutants around the photocatalyst so as to improve the degradation efficiency; (2) zeolite mineral powder can hinder the crystallite growth of photocatalysts on its surface and regulate their microstructure to influence the catalytic activity of composite photocatalysts; (3) abundant negative hydroxyl groups on the surface of zeolite could effectively improve the separation of photoinduced carriers by an electrostatic repulsion force; and (4) zeolite can act as an electron acceptor to promote

the fast separation of photoinduced e⁻/h⁺ pairs, and, thus, can produce more reactive oxygen species to accelerate the photodegradation process. Figure 1 shows a schematic diagram of the design strategies for various applications of zeolite mineral composites to remove EOPs [51].

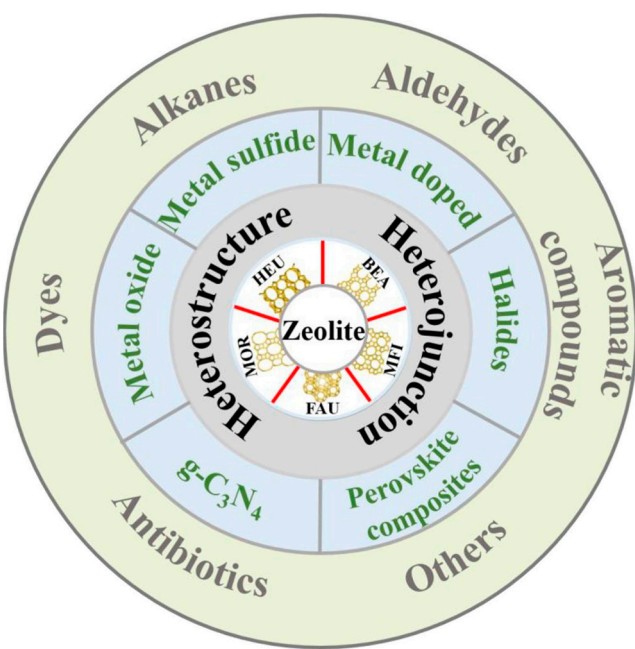

**Figure 1.** Schematic illustration of the zeolite-based composites and their environmental applications [51]. Copyright 2021 Elsevier.

As is well known, zeolite includes natural zeolite and artificial synthetic zeolite. Both natural and synthesized zeolites are characterized by a microporous crystal structure with an opening pore size smaller than 20 Å, which facilitates their wide applications as an adsorbent [52], in gas separation [53], as a water softener [54], and as a catalyst [55]. Generally, the cost of natural zeolites is usually much less than that of synthetic ones, and natural zeolites, after pretreatment, could have an equally superior performance to synthetic zeolites, which makes their use in the treatment of drinking water and watershed water very attractive [56]. Especially, as one of the most representative zeolite minerals, clinoptilolite (CP) has great potential as an adsorbent to treat wastewater, and clinoptilolite-based composite photocatalysts exhibit outstanding photocatalysis performance in degrading EOPs in waters [57]. Nezamzadeh-Ejhieh's research team had carried out a lot of pioneering work on the preparation and application of CP-based composite photocatalysts [58–60]. The applications of CP for the catalytic degradation of hazardous organics make this material highly attractive for further research. Some indications of using natural zeolites to treat large-scale municipal and industrial wastewater looks promising.

Although considerable research efforts have been carried out on the zeolite-based (natural and synthesized zeolites) photocatalyst in the past ten years (Figure 2), there are no review papers that comprehensively describe the applications of zeolite mineral (clinoptilolite) composite photocatalysts for the removal of EOPs from water streams. In this integrative review, the photocatalytic performance of semiconductor-based photocatalysts supported by CP in degrading various EOPs is summarized. Different kinds of CP-based composite photocatalysts will be introduced in the following sections. Their preparation methods and photocatalytic properties will also be introduced. The main content of this review is illustrated in Figure 3.

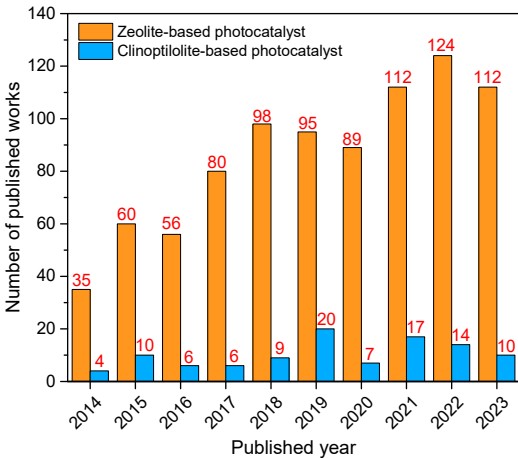

**Figure 2.** The number of publications reporting zeolite (clinoptilolite)-based photocatalyst. Source: Web of Science.

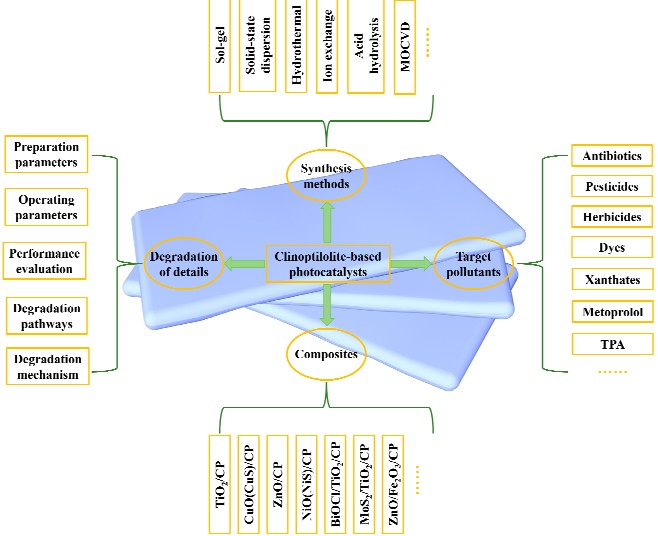

**Figure 3.** The main content of this review.

## 2. Zeolite Minerals Overview

### 2.1. Types and Structure Characteristics of Zeolite Minerals

The term zeolite was first proposed by Cronstedt in 1756, who found that these minerals could expel water, like boiling when heated [61]. They often can be found in many zeolitic sedimentary rocks and compacted deposits of volcanic ash [62]. Actually, zeolite minerals are a class of aluminosilicate-type microporous materials consisting of tetrahedral $SiO_4$ and $AlO_4$ units. The framework type of zeolites depends on the stacking and linking modes of these tetrahedral $SiO_4$ and $AlO_4$ units, which directly determines the structural characteristics of the zeolite, including the inner pore, cavity, and channel. Variations in zeolite types can be primarily attributed to their differences in chemical compositions and crystal structures. Since each $AlO_4$ tetrahedra unit produces a net negative charge, extra cations ($Na^+$, $K^+$, $Ca^{2+}$, $Mg^{2+}$, $H^+$, etc.) are indispensable to compensate, which are held loosely and, thus, can be exchangeable with other substances such as heavy metal ions [63]. The general formula of natural zeolites could be described as follows: $M_{x/n}(AlO_2)_x(SiO_2)_y \cdot zH_2O$, where M is the compensating cations of the negatively charged framework, y/x is the Si/Al ratio, and z represents the number of water molecules [64]. Zeolite minerals can be divided into three types according to the molar ratio of Si/Al or $SiO_2/Al_2O_3$, namely, low silica zeolite (Si/Al $\leq$ 2), intermediate silica zeolite (2 < Si/Al < 5), and high silica zeolite (Si/Al $\geq$ 5). Generally, low silica zeolites have

a unique advantage when acting as adsorbents in removing $NH_4^+$ or other heavy metal ions owing to their outstanding cation exchange capacity (CEC), while high silica zeolites can be used as effective organic pollutants adsorbents because the zeolites become more hydrophobic with an increase in the Si/Al ratio [65].

Up until now, over 80 types of natural zeolites have been authenticated according to the Structure Commission of the International Zeolite Association (IZA-SC) [63], while only seven kinds of zeolite minerals with abundant reserves have the utilization value to be exploited, including clinoptilolite, chabazite, ferrierite, phillipsite, mordenite, erionite, and analcime [66]. Moreover, 248 kinds of zeolite framework types were recorded by IZA-SC. Among all the zeolite types, Faujasite (FAU), mordenite framework inverted (MFI), mordenite (MOR), zeolite beta (BEA), and heulandite (HEU), known as the "Big Five" zeolites, are the most widely used in the photocatalysis process. Their detailed structural characteristics are summarized in Table 1, referring to the report of Hu et al. [51]. Especially, clinoptilolite is one of the most abundant minerals in the zeolite family [67]. It has a monoclinic framework consisting of a ten-member ring and two eight-member rings formed by the linkage of the $AlO_4$ and $SiO_4$ tetrahedral units. These two pore systems are interconnected within the lattice hosting the exchangeable cations and $H_2O$ molecules.

**Table 1.** Structural properties of five kinds of zeolite framework types. * [51] Copyright 2022 Elsevier.

| Zeolite Framework Types | Framework Structures | Maximum Diameter of a Sphere | | Typical Zeolite Materials |
| --- | --- | --- | --- | --- |
| | | That Can Be Included (Å) | That Can Diffuse along a × b × c (Å) | |
| FAU |  | 11.24 | 7.35 × 7.35 × 7.35 | Zeolite X Zeolite Y |
| MFI |  | 6.36 | 4.7 × 4.46 × 4.46 | ZSM-5 |
| MOR |  | 6.7 | 1.57 × 2.95 × 6.45 | Mordenite |
| BEA |  | 6.68 | 5.95 × 5.95 × 5.95 | Beta |
| HEU |  | 5.97 | 3.05 × 1.34 × 3.67 | Clinoptilolite |

* Data from the Database of Zeolite Structures, Copyright © 2017 Structure Commission of the International Zeolite Association (IZA-SC).

## 2.2. Physicochemical Properties of Zeolite

Because they contain abundant regular channels and pores, zeolite minerals have a large specific surface area, which is generally acceptable as promising candidates for hybrid adsorbent–photocatalyst applications. In addition, zeolite minerals possess a high adsorption ability, excellent cation exchange capacity, and abundant acid/base sites owing to their unique structural properties and chemical compositions [51]. The extraordinary physicochemical properties of zeolite minerals are summarized as follows: (1) Adsorption property: After eliminating zeolitic water by heating, the cavities and channels inside the zeolite will be "vacant"; therefore, zeolites exhibit a fairly strong adsorption force towards gases and vapors. They possess a high affinity for polar molecules such as $H_2O$ and $CO_2$ and have excellent adsorption performance under the combined action of the dispersion force and electrostatic force [68]. Zeolite minerals are widely used in industry to prepare adsorbents and desiccants for the adsorption and separation of target gases/liquids [69]. Liu et al. [70] investigated the adsorption mechanism of natural zeolite towards high-concentration $NH_4^+$ in an aqueous solution. The adsorption process was revealed by the combination of experimental optimization and molecular dynamics (MD) simulation, which demonstrates that the $NH_4^+$ adsorption process is the synergistic effect of ion exchange, electrostatic attraction, and chemisorption, as well as hydrogen bond. (2) Cation exchange property (CEC): Zeolite minerals contain many cations, including $Ca^{2+}$, $Na^+$, and $K^+$, which can be exchanged reversibly with other ions. Morante-Carballo et al. [71] summarized the scientific publications from 1970 to 2020 on the CEC of natural zeolite. The results indicate that zeolite minerals have potential applications in the field of catalysts, heavy metal removal, bioremediation, wastewater treatment, and the construction industry based on the CEC. (3) Co-catalytic performance: Zeolite minerals show good catalytic activity for various chemical reaction processes. Acid sites in the zeolite framework can catalyze multiple chemical reactions [72]. Especially, the catalytic activity of zeolite will be more prominent after depositing metal species catalysts on its surface [57]. (4) Thermal stability: Kukobat et al. [52] investigated the porosity and crystal structure properties of clinoptilolite at particular thermal treatment temperatures by using BET, FTIR, and XRD. They found that the specific surface area of clinoptilolite increased obviously after thermal treatment at 473 K because of the zeolitic water evaporation, and the monoclinic crystal structure of clinoptilolite was not changed after thermal treatment at around 873 K, indicating the structure stability of clinoptilolite. Figure 4 shows the unit cell of the monoclinic clinoptilolite crystal. Herein, Si and Al atoms are the tetrahedral atomic arrangement of Si–O and Al–O bonds, while Mg, K, Ca, and $H_2O$ occupy the pores of the clinoptilolite crystal.

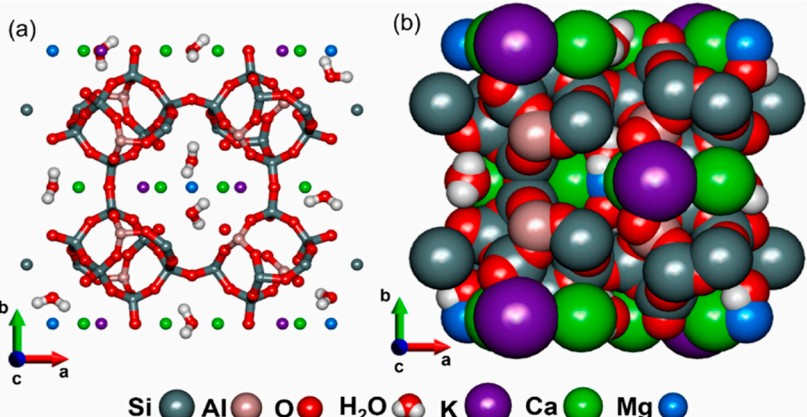

**Figure 4.** Unit cell of the clinoptilolite crystal observed in x–y direction; (**a**) clinoptilolite framework in the atomic representation, showing the pore structure at (001) face; and (**b**) clinoptilolite framework in van der Waals representation, showing the occupation of the pores with adsorbed molecules at (001) face [52]. Copyright 2022 Elsevier.

## 3. Zeolite Mineral Pretreatment

Generally, the purchased CP tuff should be crushed and sieved first, and then milled into a suitable particle size, before using it as the catalyst support. XRD, SEM, FTIR, and BET characterizations were normally carried out to analyze the crystallization, morphology, and microstructure of the CP particles. It is reported that high-energy milling can create native defects in solids, and the reduction of the particle size of CP could cause larger external surface areas to be available for interaction in adsorption and catalytic applications [73]. Moreover, it is necessary to purify the CP through acid treatment because $H_3O^+$ could replace the metal cations in the CP framework, and the undesirable water-soluble substances can be eliminated, which will unclog the tunnel and pores within the CP and, thus, increase the specific surface area, as well as the porosity.

In addition, it was found that ion exchange–alkaline treatment also exhibited a remarkable synergetic effect that could endow the CP with a more mesopore structure with a good adsorption capacity, which makes CP more suitable as a support for photocatalysts [74]. Mehrabadi and Faghihian [75] reported a combined technique of refluxing and magnetic separation to purify the CP by removing the soluble and magnetic impurities. Clearly, the pretreatment of the zeolite mineral aims to increase the active sites of zeolite support through a reduction in particle size and an improvement in the specific surface area. A typical process of CP treatments and photocatalyst preparation is shown in Figure 5.

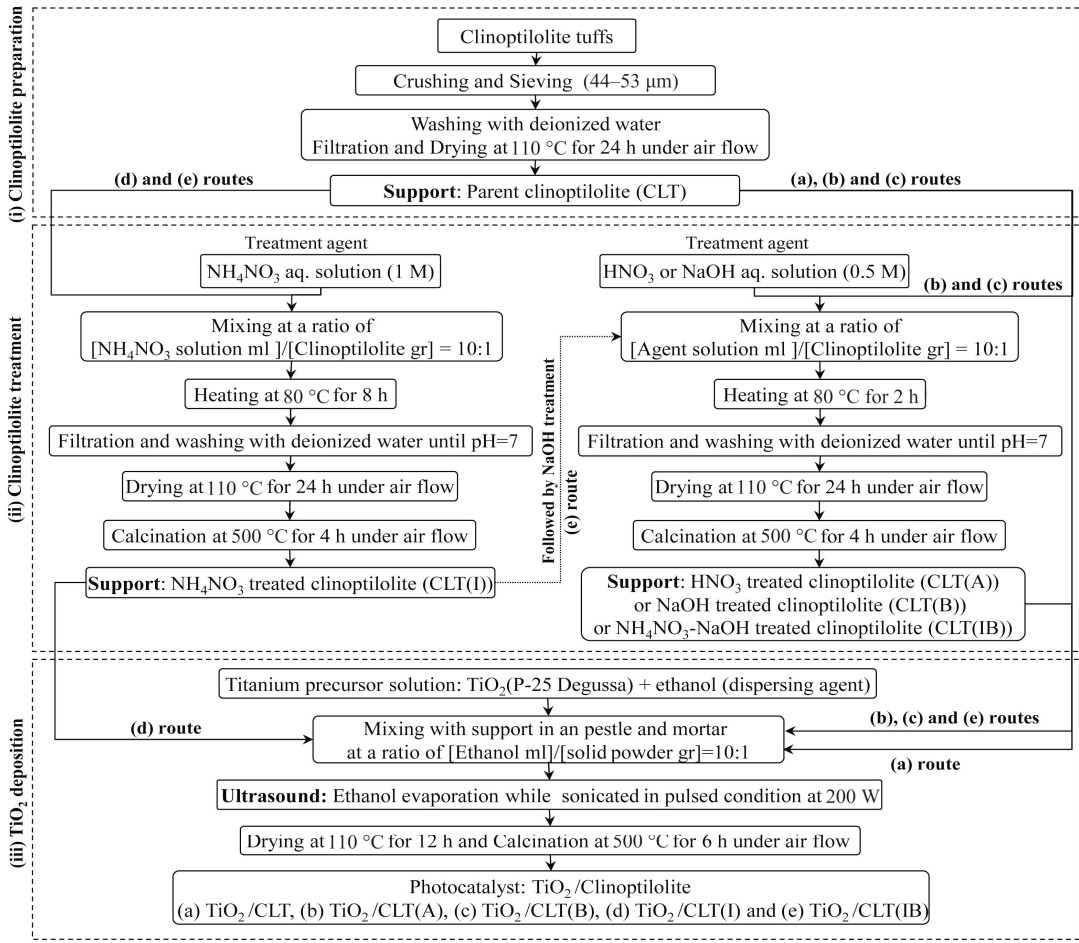

**Figure 5.** Synthesis steps of TiO$_2$/CP photocatalysts via different treatment methods [74]. Copyright 2018 Elsevier.

## 4. Common Semiconductor/Clinoptilolite Photocatalysts

There is no denying that $TiO_2$/CP is the most commonly acceptable binary clinoptilolite-based composite photocatalyst since $TiO_2$ is widely applied in the degradation of EOPs [76,77]. Li et al. [78] focused their research on the interface effect between the $TiO_2$ and clinoptilolite mineral. The characterization results showed that the $TiO_2$ NPs integrate with CP through Ti–O–Al and Ti–O–Si bonds, which demonstrates that the combination of $TiO_2$ and CP is tight via chemical force instead of simple mixing. Compared with pristine $TiO_2$, the as-prepared $TiO_2$/CP showed the higher photocatalytic activity of MO at a low initial concentration. The introduction of CP can help us avoid the passivation of $TiO_2$ in the photocatalytic degradation process. The solid-state dispersion (SSD) method was used to synthesize the $TiO_2$/CP photocatalyst, and the degradation of Acid Red 114 in water was investigated. The results indicated that the maximum degradation efficiency was obtained with the components of 10% $TiO_2$ and 90% CP under the irradiation of four mercury lamps (Philips 8W) [79].

In addition, coupling silver halides onto CP support to form AgX (X = I, Br, Cl)/CP composites can be used as photocatalysts. As another kind of common photosensitive semiconductor, silver halides can act as a reliable and efficient photocatalyst under UV–vis light when using a suitable support such as CP to scavenge the holes ($h^+$) and capture the electrons ($e^-$), which will prevent the $e^-$ from combing with $Ag^+$ ions. Moreover, the presence of CP could restrain the aggregation of AgX nanoparticles by immobilizing them onto the ion exchange sites and, thus, increase the photocatalytic activity of these Ag halides. Jafari and Nezamzadeh-Ejhieh [80] investigated the photodegradation of a mixture of 4-chloro-3-nitro aniline (CNA) and 4-methoxy aniline (MA) aqueous solution with the addition of AgX/CP photocatalysts under UV irradiation. UV–vis spectroscopy was used to monitor the degradation extent of the MA/CAN mixture pollutants, and the chemical oxygen demand (COD) was used to estimate the mineralization of the mixtures. Since ZnO can absorb a larger range of the light spectrum than $TiO_2$, researchers investigated the photocatalysis properties of ZnO/CP in different kinds of wastewater, such as bromothymol blue [81], phenylhydrazine [59], 4-nitrophenol [82], and aromatic amine [83] aqueous solutions. Mohammadzadeh et al. [84] prepared the ZnO/CP composites by mixing the $Zn(NO_3)_2$ solution and CP powder. The photocatalyst could degrade about 88% of methylene blue in 40 min under optimal conditions. Moreover, NiO, with an energy band gap of about 3.6–4 eV, can act as a promoter to produce OH radicals during the photocatalysis process. NiO/CP composite photocatalysts were prepared to degrade cefuroxime [85], cotrimoxazole [86], and 2,4-dichloroanilyne [87] under UV light. Similarly, NiS/CP photocatalysts were obtained by the same synthetic route and used to degrade furfural [88]. Both two Ni-based semiconductors have the unique advantage of being used as photocatalysts. It is also believed that $SnO_2$ can be a good photocatalyst candidate. Šuligoj et al. [89] reported the synthesis of $SnO_2$/CP by using a precipitation–deposition method. $SnO_2$ NPs with a size of 17 nm were deposited on the surface of CP without destroying its crystal structure, and the degradation of methylene blue (MB) was performed under visible light illumination. As an important narrow bandgap semiconductor (1.4 eV), CuO also displays promising application prospects in the fields of photocatalysis. CuO/CP photocatalysts are used for the photodecolorization of an aqueous mixture of cationic dye MB and anionic dye bromophenol blue (BPB), because the mixture is more similar to the real effluent. At the optimized conditions, the decolorization levels for MB and BPB, respectively, correspond to 61% and 32% within 180 min under UV irradiation [90]. Interestingly, CuS/CP composites also show the active photocatalytic property when degrading the mixture of methyl orange (MO) and bromocresol green (BCG) [91].

The above-mentioned works of literature have confirmed that binary semiconductor/CP composites can be endowed with an excellent adsorption ability and effective photocatalysis towards pollutants due to the introduction of CP support. However, almost all the binary semiconductor/CP composites still need to be stimulated to degrade the contaminants under UV light irradiation. The inert nature of the bare semiconductors

remains unchanged. The construction of a heterojunction by coupling individual semiconductors with other photosensitive ones is considered a very useful method with which to improve the visible light response ability of the bare component. Generally, the inherent shortcomings of pristine semiconductors such as $TiO_2$, including the poor adsorption ability, low visible light response, and fast photogenerated carrier recombination, could be remedied by forming the $X/TiO_2/CP$ ternary structure. Moreover, the construction of a CP-based ternary composite is also a feasible strategy used to synthesize other kinds of visible-light-driven catalysts.

Unfortunately, some ternary composite photocatalysts still need to use UV light to activate and to produce photoinduced carriers and finish the degradation of the target pollutants. Alireza Nezamzadeh-Ejhieh's group also prepared a series of samples with a semiconductor heterojunction, including $ZnO-TiO_2$ [92], $MnO-Ag_2O$ [93], $ZnS-NiS$ [94], $NiS-PbS$ (NiO-PbO) [95], $PbS-CdS$ [96], $ZnO/CuO$ ($ZnS/CuS$) [97], $ZnO-SnO_2$ [98], and $NiO-SnO_2$ [99], deposited on the CP surface. All these ternary composites were prepared by using the ion exchange method and motivated under UV irradiation. The results suggested that the rational construction of semiconductor heterojunctions is crucial for enlarging the light response range of CP-based composites.

## 5. Preparation Methods of Zeolite-Based Composite Photocatalysts

In this section, different methods are provided to prepare zeolite mineral composites for the removal of various EOPs that are present in drinking water and watershed water. Apart from the characteristics of semiconductor catalysts and supports, the synthesis method also has a significant influence on the composition and photocatalytic activity of the obtained composite photocatalysts. Up until now, multiple synthetic routes have been carried out to prepare semiconductor-based photocatalysts supported by CP such as hydrothermal [100], photochemical reduction [101], $TiCl_4$ hydrolysis [102], sol–gel [103], ion exchange [104], physical mixing [105], and SSD [106]. Herein, different methods to prepare $TiO_2/CP$ composites and their photocatalytic efficiencies in decomposing pollutants are summarized in Table 2, and typical synthetic methods are introduced as follows in detail.

**Table 2.** Different methods to prepare $TiO_2/CP$ composites and their photocatalytic efficiency in decomposing various pollutants.

| Synthesis Route | Pollutant | Posed Threat | Light Source | Optimized Degradation Conditions/Degradation Efficiency | Reference |
|---|---|---|---|---|---|
| Sol–gel | Sulfadiazine (SDZ) | Emergence of ARB and ARGs | 20 W UV lamp at 265 nm | More than 90% of SDZ (10 mg/L, 50 mL) could be removed within 120 min of both adsorption and degradation by $TiO_2/CP$ dosage of 1 g/L at neutral pH. | [107] |
| | Mixture of Methyl Orange (MO) and Methylene Blue (MB) | Water pollution | 450 W Hg lamp within the range of 300–600 nm | —— | [108] |
| | monoethanolamine (MEA) | —— | UV lamp | —— | [103] |
| Ion exchange | Mixture of aniline (AN) and 2, 4-dinitroaniline (DNAN) | Water pollution | 30 W Hg lamp | 10 ppm with respect to each pollutant, pH 5.8, catalyst dose of 0.1 g/L, and contact time of 6 h. | [109] |
| | Mixture of 2,4-dichlorophenoxyacetic acid (2,4-D) and 2-methyl-4-chlorophenoxyacetic acid (MCPA) | Water pollution | 75 W UV Hg lamp at 365.4 nm and sunlight irradiations | —— | [75] |

**Table 2.** *Cont.*

| Synthesis Route | Pollutant | Posed Threat | Light Source | Optimized Degradation Conditions/Degradation Efficiency | Reference |
|---|---|---|---|---|---|
| Hydrothermal | Xanthate | Water and soil pollution | 300 W UV irradiation | The TC-0.6 sample with $H_2O_2$ addition of 4 mL and calcination temperature of 400 °C is the most appropriate photocatalyst for the degradation of SIPX, and the degradation efficiency is 91.2 under the initial SIPX concentration of 20 mg/L and $TiO_2$/clinoptilolite dosage of 1.0 g/L within 30 min. | [100] |
| Solid-state dispersion | Azo Dye Direct Yellow 12 (DY12) | —— | 8 W Hg lamp | The maximum photodegradation efficiency was observed at 10 wt.% $TiO_2$ and 90 wt.% clinoptilolite; the optimum pH was about 2. | [106] |
| | Acid Red 114 | —— | 32 W Hg lamp | 10 wt.% $TiO_2$ and 90 wt.% clinoptilolite | [79] |
| | Reactive Black 5 (RB5) | —— | 8 W UV lamp | 10 wt.% $TiO_2$ and 90 wt.% clinoptilolite | [110] |
| $TiO_2$-impregnation | 2,4-dichlorophenoxyacetic acid (2,4-D) | Kidney and liver problems and carcinogenic effects on the animals and human | 60 W Hg lamp and visible light irradiations | The optimized degradation efficiency was obtained at pH = 6, catalyst dose of 0.4 g/L, $TiO_2$ loading of 5%, and pollutant concentration of 6 mg/L; the degradation efficiencies, corrspond to 58.0 and 31.0% under UV and visible light irradiation. | [111] |
| Physical mixing | Metoprolol | —— | UV lamp | Photodegradation efficiency higher than 95%. | [105] |
| Acid hydrolysis | Terephthalic acid (TPA) | Acute, chronic, and molecular toxicity to organisms | 8 W UVC lamp at 254 nm | —— | [102] |
| MOCVD | Salicylic acid (SA) | —— | 125 W UV irradiation within the range of 200–280 nm | —— | [112] |

——Not Applicable.

### 5.1. Sol–Gel Method

Sol–gel chemistry is widely used to synthesize nanocomposite materials due to the easy conversions of a wide variety of precursors and the very mild reaction conditions (near room temperature). The sol–gel process usually contains the evolution of inorganic networks via forming a colloidal suspension (sol) and gelling the sol to generate a continuous network in the liquid phase (gel) [113]. Taking the preparation of $TiO_2$ as an example, metal alkoxides such as titanium tetraisopropoxide (TTIP) and titanium butoxide (TBT) are often used as precursors. The reaction equation is written as follows [114]:

(1) Hydrolysis:

$$Ti - OR + H_2O \rightarrow Ti - OH + R - OH \tag{1}$$

(2) Condensation:

$$Ti - OH + RO - Ti \rightarrow Ti - O - Ti + R - OH \tag{2}$$

$$Ti - OH + HO - Ti \rightarrow Ti - O - Ti + H_2O \tag{3}$$

The sol–gel method is considered to be a suitable technique to deposit semiconductor metal oxides such as $TiO_2$ onto the zeolite mineral surface. After adding zeolite minerals into the reaction system during sol formation, the hydrolysis rate can be controlled because of the unique microporous structure of zeolite, which offers the adsorption and sustained release of water, and, thus, restrain the aggregation of the $TiO_2$ sol to form large NPs. Moreover, $TiO_2$/CP composites prepared by sol–gel chemistry show strong adhesion between $TiO_2$ and clinoptilolite through chemical bonding. Khodadoust et al. [103] utilized the sol–gel method to prepare $TiO_2$/CP, and the composite photocatalyst was used to degrade monoethanolamine (MEA), which is used in the gas and oil refining process. The gathering of MEA in wastewater could also cause negative effects on natural ecosystems and human health.

## 5.2. Ion Exchange Method

The typical ion exchange properties of zeolite minerals derive from easily exchanging $Ca^{2+}$, $Na^+$, and $K^+$ with many other metal cations. This method has been developed to stabilize the nano-scale metal on the CP surface. The research group of Alireza Nezamzadeh-Ejhieh reported the synthesis of metal oxide semiconductor/clinoptilolite (MOS/CP) composites by ion exchange with the presence of different kinds of metal ion aqueous solutions, including $Cu^{2+}$, $Zn^{2+}$, $Ni^{2+}$, and $Fe^{2+}$ solutions, and the obtained composites are treated by a calcination process. They focused on studying various transition metal oxides/sulfides incorporated into the CP, and all the obtained composites, including CuO/CP [90], CuS/CP [91], ZnO/CP [82], NiO/CP [85], and NiS/CP [88], exhibited superior photodegradation activity to multiple pollutants. Moreover, all these semiconductors are widely available, inexpensive, and non-toxic. The ion-exchanging method is also applied to prepare the FeO/CP composite. The obtained samples showed good photodegradation performance for the cotrimoxazole [115], phenol [116], furfural [117], 2,4-dichloroaniline [58], and one of the most commonly detected antibiotics, namely, tetracycline [60], in wastewater.

## 5.3. Hydrothermal Method

The hydrothermal method is generally conducted in a high-pressure autoclave or specially sealed container under temperatures ranging from 100 to 250 °C. It refers to the synthesis of inorganic materials through a chemical reaction in an aqueous solution above the boiling point of water [118]. The expected morphologies and sizes of MOS can be obtained and anchored on the zeolite surface by changing the hydrothermal temperature and time in this closed system. Shen et al. [100] prepared $TiO_2$/CP composites using acid-leaching CP as support via a hydrothermal method. From the TEM image, it can be observed that $TiO_2$ NPs were coated on the CP surface with good dispersion. Moreover, the average particle size of $TiO_2$ NPs was about 20 nm, which can endow the composites with more excellent photocatalysis. Taking sodium isopropyl xanthate (SIPX) as the target pollutant, the photodegradation test results showed that over 90% of SIPX could be eliminated within 30 min with a composite dosage of 1.0 g/L under neutral conditions.

## 5.4. Ex-Situ Methods

Herein, ex-situ methods are primarily defined to distinguish them from in-situ methods. Common ex-situ methods to prepare zeolite mineral composites involve SSD, $TiO_2$-impregnation, and physical mixing. Hybrid $TiO_2$/CP photocatalysts were prepared by the SSD route and embedded in the outer layer of dual-layer hollow fibre (DLHF) membranes. The inner layer of DLHF plays a vital role in the separation process, while the outer layer consisting of $TiO_2$/CP is used to degrade Reactive Black 5 (RB5) under UV light. The $TiO_2$/CP composite with the components of 10 wt.% $TiO_2$ and 90 wt.% CP showed higher RB5 photodegradation efficiency in suspension conditions [110]. Mehrabadi and Faghihian [111] reported the preparation of $TiO_2$/CP composites by impregnating thet CP mineral with $TiO_2$ and analyzed their photodegradation activity towards dichlorophenoxyacetic acid (2,4-D) under UV and visible light irradiation. The degradation efficiency corresponds to 58.0 and 31.0% under different irradiation sources. Farzaneh et al. [119] used the Seman natural zeolite (clinoptilolite) and $TiO_2$ P25 as materials to prepare a mixed photocatalyst through stirring and ultrasonication. The photodegradation performance of tetracycline (TC) in an aqueous solution was studied using the as-prepared $TiO_2$/CP. The results showed that a maximum degradation of TC (more than 77%) was obtained at the optimum conditions of pH 5.9, a catalyst concentration of 0.3 g/L, and a TC concentration of 8 mg/L within 60 min of irradiation. A physical mixing method was used to synthesize the $TiO_2$/CP composite to achieve the synergistic effect between the $TiO_2$ and Mexican clinoptilolite. Metoprolol, a beta-blocker medicament, is usually applied to treat arrhythmia, hypertension, and heart failure. $TiO_2$/CP exhibited higher photocatalytic

activity towards metoprolol than the pristine $TiO_2$ due to the positive synergistic effect between the $TiO_2$ catalyst and the clinoptilolite support [105].

In general, these ex-situ methods are easily scalable to prepare a large amount of zeolite mineral composites. However, the semiconductor dispersed on the surface of zeolite minerals is random and stacked. Moreover, the interaction between semiconductors and zeolites is usually weak. All these conditions are disadvantageous for using the composites as photocatalysts. The main difference between in situ and ex situ methods is concentrated on the reaction mechanism. For in-situ methods, the composites are obtained in the reaction of chemicals, while, for ex situ methods such as SSD, the object semiconductor and zeolite mineral are contacted at the interface without chemical bonding. Actually, each preparation method possesses its own advantage and limitation. It is unscientific to assert the most appropriate preparation method. Researchers should rationally design the CP-based composites based on the structure–activity relationships between the CP surface properties and catalysts microstructure, as well as the photocatalytic activity of the composites. Importantly, the influence of the fabrication cost and photocatalytic activity of the composite photocatalyst should be comprehensively assessed before putting it into application.

## 6. Degradation of Emerging Organic Pollutants

Although most of the EOPs are not included in the current water quality legislation, their existence in low concentrations can lead to serious consequences in human health and the environmental ecosystems [77,120]. Especially, EOPs such as PPCPs, perfluorinated compounds (PFCs), antibiotics, EDCs, and MPs have been detected at varying levels in domestic wastewater, industrial wastewater, aquaculture wastewater, agricultural wastewater, hospital wastewater, and other water bodies. As we all know, EOPs are the substances produced by industrial, agricultural, and household activities, which will be discharged into the aquatic environment and end up in the marine environment, as shown in Figure 6. The water environment is the main carrier of EOPs; more than 90% of EOPs will enter natural water bodies [121]. In the water circulation system, EOPs enter surface water and groundwater through various channels such as runoff, diffusion, and infiltration, causing the pollution of drinking water sources and potential threats to aquatic organisms, ecological security, and human health. Wu et al. [122] focused their investigations on reducing the disease burden induced by EOPs in water. They not only drew the spatial distribution of typical EOPs in main river basins in China (Figure 7a), but also assessed the human exposure risks to waterborne EOPs and found the relationship between them and carcinogenic risks (Figure 7b), which is very valuable for reference in order to establish and implement feasible management strategies to prevent diseases. Notably, these wastewaters enter the sewage pipe network without effective treatment and eventually enter the urban sewage treatment plant, which becomes the convergence of EOPs. Unfortunately, EOPs cannot be completely removed by the traditional treatment technique in the urban sewage treatment plant. It is important to develop green, efficient, and practical removal technologies for organic pollutants. Semiconductor-based photocatalysis technology is an efficient sewage treatment technique that has emerged and has been widely accepted in recent years. In this section, various zeolite mineral composites for the degradation of representative EOPs are summarized.

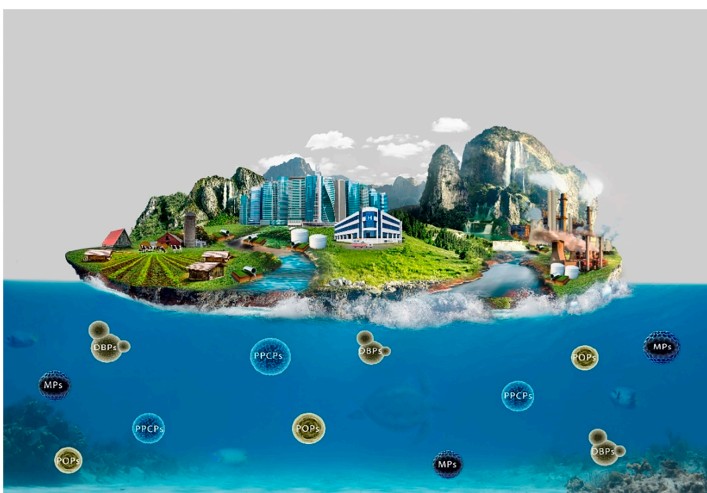

**Figure 6.** Schematic diagram of the EOPs flow in the aqueous environment [121]. Copyright 2022 Elsevier.

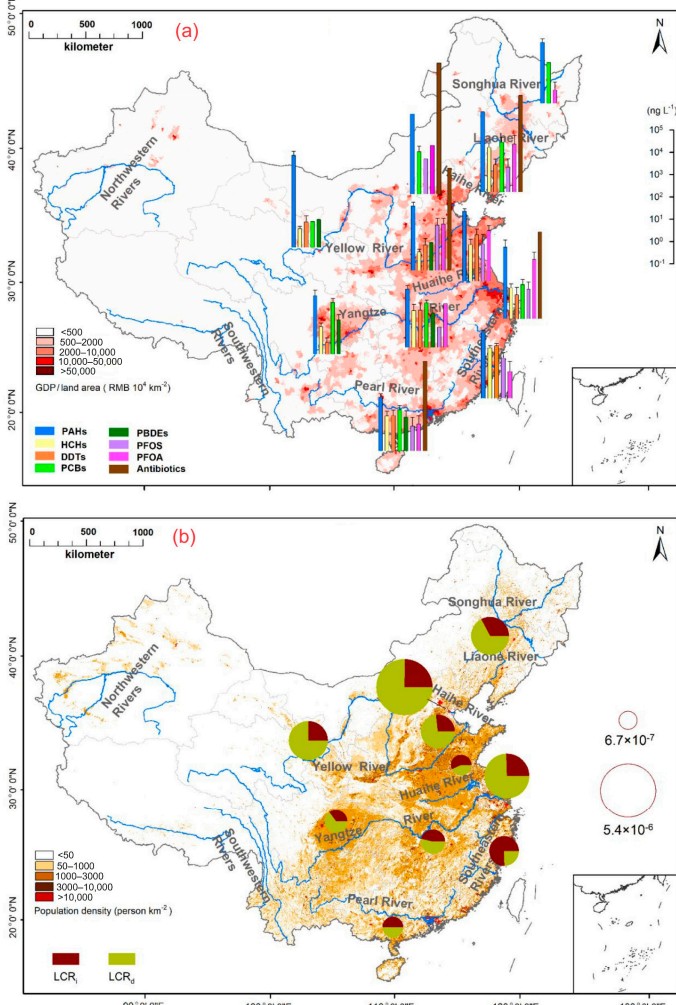

**Figure 7.** (**a**) Spatial distribution of typical organic contaminants in main river basins in China. (**b**) Sum of human lifetime cancer exposure risks of typical carcinogenic organic contaminants in water through ingestion (LCR$_i$) and dermal contact for bathing (LCR$_d$) in major river basins of China [122]. Copyright 2022 American Chemical Society.

### 6.1. Degradation of Antibiotics

As common pharmaceutical contaminants, antibiotics are broadly used to treat microbial infections in both humans and animals [123,124]. Presently, over 100,000 tons of antibiotics are consumed every year all around the world [125]. The overuse and misuse of antibiotics stimulated the more rapid emergence of antibiotic-resistant bacteria (ARB) and antibiotic-resistant genes (ARGs), which inevitably reduces their therapeutic potential against human and animal pathogens [126]. Antimicrobial resistance has been listed as a global public health crisis by the World Health Organization (WHO), that must be treated with the utmost urgency [127]. The excessive use of antimicrobial agents causes an increase in antibiotic residues in wastewater and natural water, posing serious health risks and environmental issues [128]. Antibiotic-containing effluents can enter municipal wastewater via household and animal waste. It has been proven that trace levels of antibiotic residues in water and soil can also pose a serious threat to human health through the food chain [129]. However, conventional treatments can hardly eliminate antibiotic residues from effluents. Adsorption is considered an efficient method for the removal of antibiotics, while this operation cannot achieve the complete mineralized decomposition of antibiotics and the adsorbent-containing antibiotics may cause secondary pollution to the environment.

Sulfadiazine (SDZ) is one of the most common antibiotics, which is mainly used for the treatment of toxoplasmosis, otitis media, and rheumatic fever. Little residues of SDZ in the sewage effluents of hospitals can pose serious threats to the ecosystem and human health [130]. Liu et al. [107] prepared a $TiO_2$/CP composite photocatalyst via the sol–gel method, which exhibited significantly improved photodegradation efficiency towards SDZ under UV light irradiation. $TiO_2$ NPs with a size of about 50 nm demonstrate granular dispersion on the clinoptilolite surface. The Ti–O–Si bond formed between $TiO_2$ and the clinoptilolite support accounts for the excellent photocatalytic activity, easy separation, and good stability. Based on previous research and the detection of high-performance liquid chromatography–mass spectrometry (HPLC-MS), Figure 8a provides the four possible pathways and the corresponding intermediate products in the photodegradation of SDZ under UV light irradiation. Moreover, Figure 8b further reveals the photocatalytic mechanism of SDZ in the presence of $TiO_2$/CP under UV light. Obviously, all the by-products will be mineralized into $CO_2$ and $H_2O$ to achieve the complete removal of SDZ in the wastewater.

Tetracycline (TC) has been widely used as a pharmaceutical drug for therapeutic purposes and has also been applied as a feed additive to treat animal diseases, thus facilitating the development of animal husbandry and fish farming due to their excellent antimicrobial activity and cost-effectiveness. The massive emission of wastewater containing TC antibiotics caused a serious threat to the environment, ecosystem, and public health owing to their degradation resistance, long half-life, and poor metabolism [131,132]. An Ag/g-$C_3N_4$/CP composite with high photoactivity was prepared by ultrasound energy and was used to remove the TC under simulated solar light irradiation [133]. DRS analysis demonstrated that the deposition of Ag NPs could improve the absorption intensity in the simulated solar light region owing to the surface plasmon resonance, and decrease the bandgap resulting in more solar spectrum utilization. About 90% of the TC in the aqueous solution could be degraded within 3 h. Liu et al. [134] reported the preparation of $MoS_2$/zeolite (clinoptilolite) composites by combining ultrasonic and hydrothermal methods and studied their application for the degradation of TC. $MoS_2$@Z-5 exhibited the highest efficiency of 87.23% for TC degradation after 3 h of illumination under visible light. Ultra-performance liquid chromatography–mass spectrometry (UPLC-MS) was performed to analyze the photodegradation intermediates of TC over the $MoS_2$@Z-5 composite at different illumination times. The possible degradation pathways for TC by $MoS_2$@Z-5 are shown in Figure 9a. Although the detailed photodegradation pathways are different, the result remains the same; namely, the carbon ring of all the intermediates will be destroyed to decompose into small molecules and, finally, transform into $H_2O$ and $CO_2$ in the reaction system. Figure 9b illustrates the proposed degradation mechanism of $MoS_2$@Z-5 for TC under visible light.

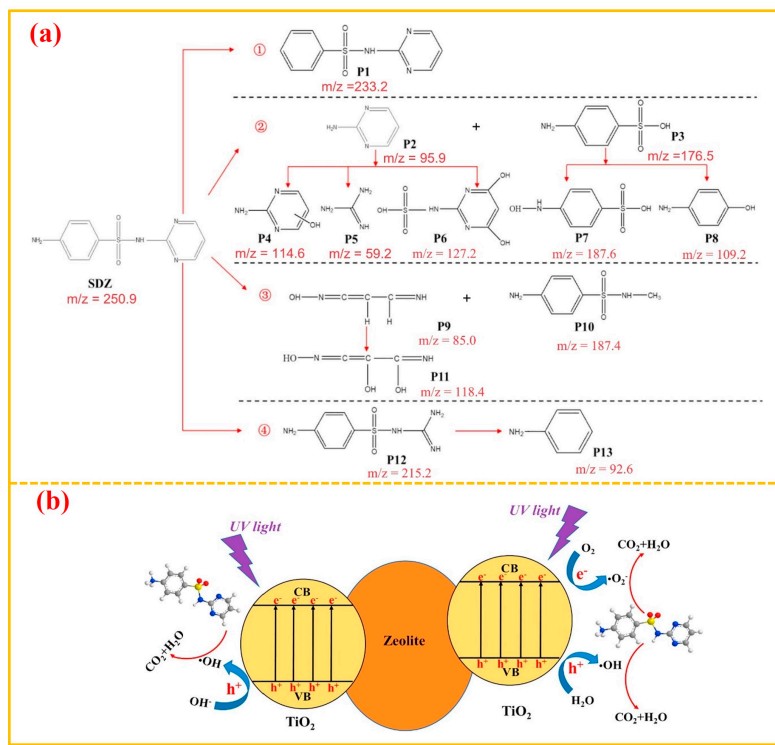

**Figure 8.** (**a**) Possible degradation pathways of SDZ by TiO$_2$/ZEO. (**b**) General photocatalytic mechanism of SDZ in the TiO$_2$/ZEO + UV system [107]. Copyright 2018 Elsevier.

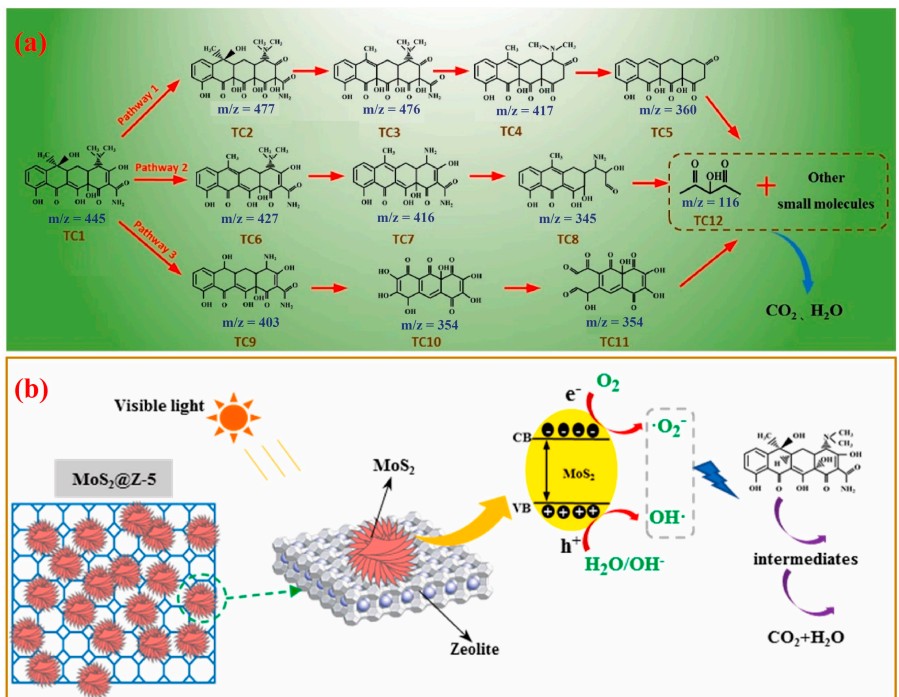

**Figure 9.** (**a**) Possible photodegradation pathways of TC in the reaction system. (**b**) Proposed mechanism of MoS$_2$@Z-5 for tetracycline photodegradation [134]. Copyright 2022 Elsevier.

Amoxicillin (AMX), common in human and veterinary medicine, belongs to a β-lactam antibiotic and is often applied for the treatment of bacterial infections, including pneumonia, urinary tract infections, and skin infections. More than 80% of AMX will be excreted through urine from human and animal bodies within 2 h of ingestion [135,136]. Therefore, excretion from humans and animals is the main source of AMX in aquatic ecosystems.

Meanwhile, the effluents produced from hospital and domestic sewage, pharmaceutical industries, and leachates from landfill sites also contain a massive dose of AMX, as shown in Figure 10. It has emerged as one of the most frequently occurring antibiotics in wastewater. However, it has been proven that wastewater treatment plants (WWTPs) cannot remove AMX thoroughly due to their solubility, biodegradability, and stability [137]. Kanakaraju et al. [138] investigated the preparation of integrated photocatalytic adsorbents (IPAs) by coupling the $TiO_2$ on the surface of natural zeolite (clinoptilolite). After an acid–alkaline pre-treatment and calcination at 300 °C, the obtained $TiO_2$/zeolite composites showed the best performance when used in the degradation of AMX. The superior performance of the IPA material for AMX degradation can be summarized as follows: (1) the excellent adsorption capability of zeolite can attract the AMX molecules to the surroundings of the composite to accelerate AMX degradation; (2) the acidic nature contributes to the hydrolysis of AMX; and (3) $TiO_2$ functions as a photocatalyst to efficiently degrade the AMX and its by-products.

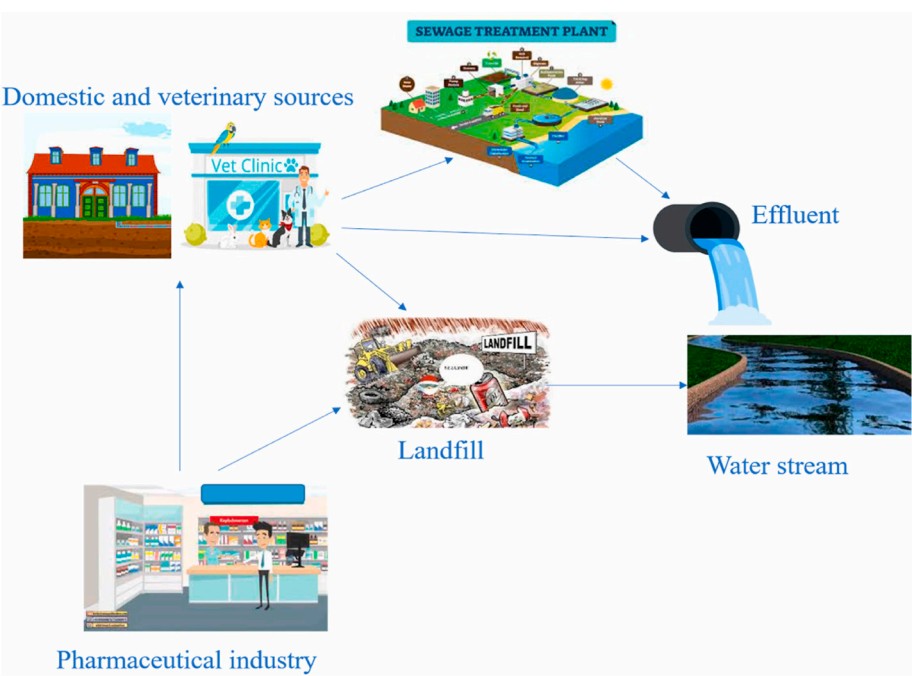

**Figure 10.** Exposure routes of AMX into surface water [136]. Copyright 2022 Elsevier.

### 6.2. Degradation of Pesticides

As we all know, pesticides are widely used to control and kill pests, which are also the main sources of water pollution. Pesticide residues exist globally in water streams and shallow water [139]. The molecular structure of pesticide usually consists of two parts: active and inert. It is the active part of a pesticide that performs the actual function to kill or repel pests, and the active ingredient of most pesticides resists environmental degradation [140]. Aniline (AN) and 2,4-dinitroaniline (DNAN) are often used to synthesize pesticides. Zabihi-Mobarakeh and Nezamzadeh-Ejhieh [109] reported the preparation of a $TiO_2$/CP photocatalyst used to degrade a mixture of AN and DNAN aqueous solution. Hybrid organics are proposed as target pollutants because they are the common by-products of a real effluent sample. The $TiO_2$/CP photocatalyst was synthesized through two steps, namely, the incorporation of Ti into CP by using ion exchange, and then calcination to form the composites.

### 6.3. Degradation of Herbicides

Herbicides are recurring pollutants in the aquatic system because of their widespread usage in the agriculture field for weed control. According to the actual effects, herbicides

can be divided into two categories of selectivity and non-selectivity. Selective herbicides can inhibit or kill specific weeds while keeping the desired crops safe. But, for non-selective herbicides, almost all plant species will be harmed [141]. The increasing and intensive use of herbicides in the past few decades has maintained the quality and quantity of agricultural crops with the continuous rise in world food production. It is supposed that over 950,000 tons of herbicides are used all around the world [142]. However, herbicides may enter the aquatic ecosystems via leaching, soil erosion, and surface runoff. Although the detection of herbicide residues and their metabolites in aquatic bodies (groundwaters and surface waters) is usually at low contents (from ng/L to μg/L), they are considered as priority persistent organic pollutants (POPs) owing to the fact that their bioaccumulation and high toxicity can affect ecosystems and human health [143].

Two phenoxy acid herbicides are 2,4-dichlorophenoxyacetic acid (2,4-D) and 2-methyl-4-chlorophenoxyacetic acid (MCPA). They are widely utilized to control various kinds of broad-leaved weeds such as cotton, tobacco, and sugarcane, and, thus, increase agricultural productivity. The large consumption of these two herbicides can be ascribed to their good selectivity, low cost, effective function, and simple availability. Mehrabadi and Faghihian [75] prepared the $TiO_2$/CP composite by shaking the mixture of purified CP and $(NH_4)_2[TiO(C_2O_4)_2]$ solution at room temperature for 8 h and annealing at 450 °C for 10 h. The total organic carbon (TOC) values were measured to evaluate the degree of mineralization of the 2,4-D, and MCPA. The results demonstrated that the deposition of $TiO_2$ on the CP surface could significantly improve the photocatalytic efficiency of pristine $TiO_2$. The HPLC technique was used to determine the degradation products of 2,4-D and MCPA.

### 6.4. Degradation of Dyes

It is estimated that over 700,000 tons of dyes are produced annually worldwide, and most of these dyes are consumed in the dyeing, textile, plastics, cosmetics, pharmaceutical, and photographic industries. These industries consume over 100,000 types of commercial synthetic dyes generated from petroleum intermediates or coal tar. Especially, since the dyeing process is inefficient, a mass of non-biodegradable dyes generated in textile industries will be discharged into the natural water source [144]. Dye that exists in industrial wastewater is mainly composed of toxic and carcinogenic substances such as aromatic amine and benzidine, which are very difficult to degrade and, thus, cause a long-term threat to human beings and the marine ecosystem. Therefore, it is crucial to treat the dye effluents before discharging them into natural water streams [145]. Figure 11 illustrates the negative effects of dyes on human health and the aquatic environment.

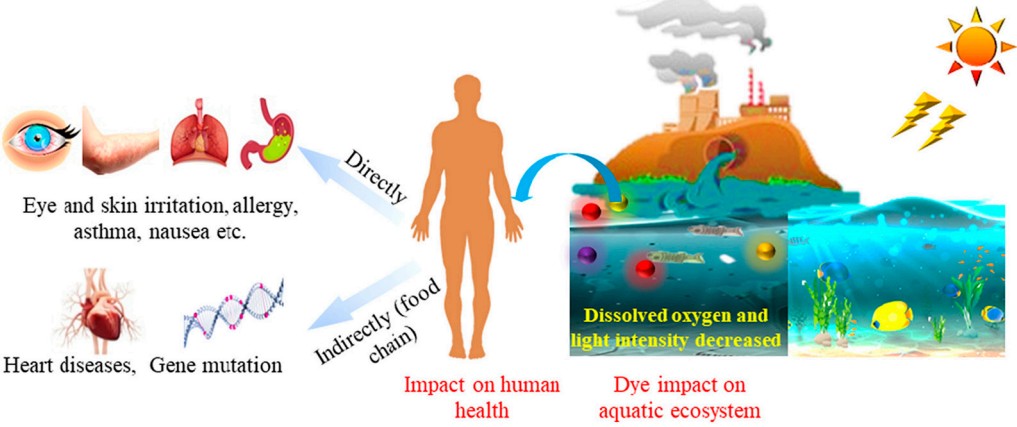

**Figure 11.** Dye impact on human health and the aquatic environment [144]. Copyright 2022 Elsevier.

Ullah et al. [146] used the hydrothermal method to prepare $TiO_2$/CP composites, and their photocatalytic performance towards crystal violet (CV) dye was studied. The

result showed that 89% of CV could be degraded within 100 min. $TiCl_4$ was chosen as the titanium source, and the effects of various parameters such as the acidity, temperature, and concentration of $TiCl_4$ on the structural properties were assessed to elucidate the synthesis mechanism of $TiO_2$/CP. Ullah et al. [147] reported the preparation of $TiO_2$/CP by using three different methods, including hydrothermal, sol–gel, and in situ hydrothermal, to investigate the relationships between the synthesis routes and physicochemical properties of the obtained $TiO_2$/CP composite photocatalysts. The photodegradation properties of different samples towards CV dye in an aqueous solution were investigated under UV light. The results showed that the in situ hydrothermal method could result in $TiO_2$ NPs on the leaf-like clinoptilolite surface with better dispersity, and the preparation of $TiO_2$/CP via the in situ hydrothermal method showed a 98% removal of CV dye. Furthermore, the sol–gel route was conducted to synthesize the $TiO_2$/CP photocatalyst, and the degradation of MB and MO mixtures was investigated under UV light irradiation; 94% MB and 83% MO could be degraded within 180 min. Moreover, the photocatalyst demonstrated excellent reusability and stability for practical applications. After crushing and sieving, the clinoptilolite particles with a size of less than 100 μm were dealuminated by the ion exchange method [108]. Moreover, ternary $MnO/Ag_2O$/clinoptilolite (MAC) composites were prepared by ion exchange, and the as-obtained samples were used for the phodecolorization of MB aqueous solution [93]. Tan et al. [148] prepared a novel $BiOCl/TiO_2$/clinoptilolite (BTC) composite with a "sheet–particle–sheet" structure by homogeneous precipitation and the calcination–crystallization process (Figure 12). The composite showed more active photocatalysis towards rhodamine B (RhB) than pristine $TiO_2$ and BiOCl under visible light irradiation. The degradation kinetic constant of BTC is about 48, 11, and 30 times higher than that of the single $TiO_2$, BiOCl, and $TiO_2$/CP composite, respectively. The active radical experiment was conducted to investigate the photocatalytic mechanism of BTC, and the results showed that hydroxyl radicals ($\cdot$OH) and holes ($h^+$) are the dominant reactive species in the degradation of RhB. Moreover, tetracycline was further employed as the target pollutant, and the degradation process demonstrated the universality and non-selectivity of the present BTC photocatalyst for the various contaminants.

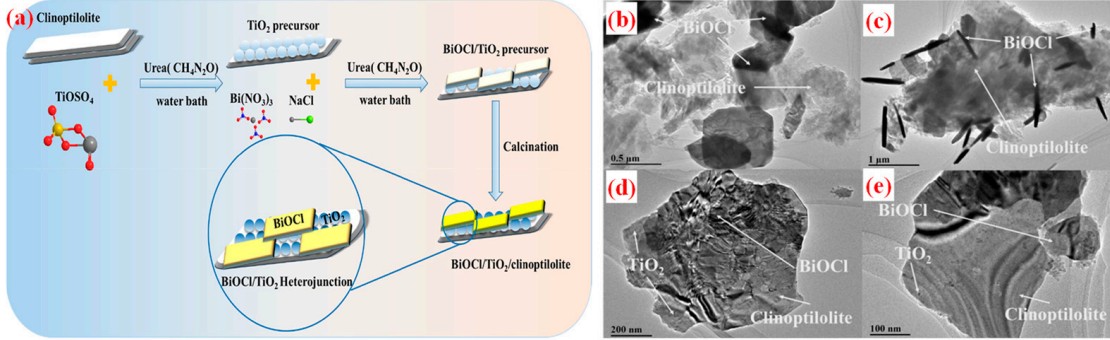

**Figure 12.** (**a**) Schematic illustration of the synthesis strategy of $BiOCl/TiO_2$/clinoptilolite composite. (**b**–**e**) TEM images of BiOCl/clinoptilolite and $BiOCl/TiO_2$/clinoptilolite [148]. Copyright 2020 Elsevier.

### *6.5. Degradation of Xanthates*

Xanthates are considered the most widely used reagents in the flotation of non-ferrous metals. Although most of the xanthates enter the concentrate product with foam, a large amount of xanthate residues remains in the slurry, which can give rise to odor nuisances and toxicity to the biota, thus leading to the deterioration of the surrounding water quality of the mine. Many flotation reagents, especially xanthates, remain in beneficiation wastewater, which will enter the tailing pond and gradually infiltrate into the natural environment. Xanthate residues in flotation wastewater can severely pollute the mining area's ecosystem and injure the human liver and nervous systems. It is important to effectively treat low-concentration and high-toxicity xanthate effluent in the construction of green mines [149–151].

Our research group [152–154] synthesized a BiOCl/TiO$_2$/clinoptilolite (BTC) composite by a simple hydrothermal route combined with a water bath precipitation procedure; BiOCl with a sheet structure and TiO$_2$ NPs were uniformly deposited on the surface of flake-like CP. SEM and TEM characterization analysis of the BTC showed the tight combination among BiOCl nanosheets, TiO$_2$ NPs, and lamellar CP instead of simple physical mixing. BTC exhibited excellent degradation efficiency to different kinds of xanthates, including sodium ethyl xanthate (SEX), sodium butyl xanthate (SBX), sodium isopropyl xanthate (SIPX), and sodium isoamyl xanthate (SIAX). Interestingly, the degradation kinetic constant increases with an increase in the xanthate formula weight. Radical scavenger tests suggested that the superoxide radical and h$^+$ were the dominant active species in the degradation reaction system. Moreover, MoS$_2$/TiO$_2$/clinoptilolite (MTC) heterogeneous photocatalysts with a hierarchical structure were prepared by a two-step hydrothermal route. Flower-like MoS$_2$ grew on the TiO$_2$/CP catalyst surface after the final "Ostwald ripening" process. The novel MTC composite showed remarkably enhanced photocatalytic activity towards xanthates under visible light illumination, which can be ascribed to the synergistic effect of CP support and MoS$_2$/TiO$_2$ heterojunction. Moreover, a Ag/TiO$_2$/clinoptilolite (ATC) composite was applied to degrade xanthates under visible light irradiation, which was prepared by combining the hydrothermal and in situ reduction process. ATC exhibited outstanding photodegradation properties compared with binary Ag/TiO$_2$ and Ag/CP composites. The deposition of Ag NPs on the surface of TiO$_2$/CP contributed to the separation of photoinduced carriers and the enhancement of visible light absorption ability. Superoxide radicals and holes played a dominant role in the photodegradation reaction system.

*6.6. Degradation of Other Emerging Organic Pollutants*

Heidari et al. [155] reported a detailed investigation on the photodegradation of the drug furosemide (FRS) using ZnO supported on ion-exchanged CP. The ZnO/CP composites were obtained via the sonoprecipitation method. FRS, a kind of prescribed pharmaceutical, is employed as the degradation object at present because of its extensive use in the treatment of edema and hypertension, and about 90% of FRS is directly released into the environment, which exhibits carcinogenic effects through food chains on human bodies. Moreover, conventional WWTPs cannot effectively eliminate FRS, and, thus, it is often found in the effluents and surface waters of sewage treatment plants. The degradation experiment results revealed that ZnO(15 wt.%)/CP showed the best photocatalytic performance towards FRS when the pH value was about 7. CuO/CP composites were used to degrade p-aminophenol (PAP) under sunlight irradiation, which is known as a kind of aromatic amine and is widely used in the textile industry. The random discharge of PAP into the environment will cause potential threats to human health, such as heart shock and mutagen [104]. Moreover, benzene-1,2-diamine (BD) [156], 2,6-dimethyl phenol (DMP) [157], and dichloroaniline (DCA) [158] were chosen as target pollutants to study the photocatalysis properties of the CuO/CP. All these organics are not easy to decompose and will also cause a series of environmental problems. Yener et al. [102] deposited rutile TiO$_2$ on the CP surface to prepare the TiO$_2$/CP composite photocatalysts through the hydrolysis of TiCl$_4$ without calcination. The photocatalytic properties of TiO$_2$/CP were investigated in the degradation of terephthalic acid (TPA), which is produced from the fabrication of polyester fibers and films and could cause serious toxicity to organisms. The present composites with larger surface areas exhibited better photocatalytic activity towards TPA under UVC illumination than the commercial Degussa P25 and anatase. Elghniji et al. [112] reported the synthesis of TiO$_2$/CP by the metal–organic chemical vapour deposition method (MOCVD) for the first time. The photocatalysis activity of TiO$_2$/CP composites was determined by measuring the degradation efficiency of salicylic acid (SA) under UV light irradiation. The results showed that the SA degradation rate of the present sample is twice faster than that of the sample obtained by the impregnation technique and commercial TiO$_2$ (P25), which could be ascribed to the homogeneous distribution of TiO$_2$ NPs on the CP support and the intimate contact between the TiO$_2$ and CP. ZnS and SnS$_2$

semiconductors coupled on the surface of CP were prepared via ion exchange, followed by sulfidation [159]. The obtained ternary composite showed the effective photodegradation performance of the phenol-containing oil refinery wastewater under sunlight irradiation. The mole ratio of ZnS and SnS$_2$ is 1.85 when the composite showed the best photocatalytic activity. The direct Z-scheme mechanism was used to illustrate the degradation process, and the results indicated that phenol could be degraded faster by the oxidation pathway than by its reduction. The experimental parameters' simultaneous effects were optimized using the response surface methodology (RSM). The value of R$^2$ proved that the data predicted by the RSM is consistent with the experimental results.

In summary, zeolite-based composite photocatalysts are widely used in the field of photodegradation EOPs. Despite the fact that researchers have obtained promising results on the lab scale, it is still a significant challenge to meet the actual requirements because real wastewater contains multiple EOPs and other organic and inorganic contaminants. Inevitably, there exists competitive adsorption over the surface of zeolite-based composites among various pollutants, which would affect the photodegradation performance towards target pollutants. Therefore, the EOP photodegradation efficiencies of the above-mentioned zeolite-based composites are questionable when used in the treatment of real wastewater. It is very necessary to carry out confirmatory tests to ensure the reliability of the degradation performance of composite photocatalysts.

## 7. Photocatalytic Performance Evaluation of the Composite Photocatalysts

Generally, the dark reaction process is necessary in order to realize the adsorption/desorption equilibrium before irradiation. The dark adsorption time depends on the character of the CP-based nanocomposites. The contaminant removal efficiency (η) of adsorption and photocatalysis could be calculated according to the following equation: $\eta = \frac{C_0 - C_t}{C_0} = \frac{A_0 - A_t}{A_0} \times 100\%$ [154,160]. C$_0$ and C$_t$ correspond to the starting concentration and the concentration at a specific degradation time, respectively, while A$_0$ and A$_t$ are the initial absorbance and the absorbance at the reaction time t. Moreover, much work so far has shown that the degradation process of the CP-based nanocomposites towards organic pollutants follows pseudo-first-order kinetics. The formula used for fitting is often written as $\ln(C_0/C_t) = kt$ [161,162], where k is the reaction rate constant, and it can be determined by the slope of the line plotting.

Researchers have studied the effects of key preparation parameters, such as the loading amounts of the semiconductor on the surface of the CP support and the calcination temperature. The loading amount of the photoactive semiconductor is closely related to the morphology and photocatalysis properties of the CP-based catalyst [79]. Meanwhile, it is found that the calcination temperature could strongly influence the crystalline phase and crystallite size of the catalysts and, thus, has a direct impact on their degradation activity [163]. Moreover, the effect of some operating parameters, including the catalyst dosage and initial concentration of the pollutant, should also be studied. All these parameters have significant influences on the adsorption and photodegradation performance of the composite photocatalyst. Theoretically, the photocatalytic efficiency will improve with an increase in the dosage, which results from more produced reactive radicals under the same light intensity. Taking the degradation rate and the application cost into consideration, an appropriate dosage with the fastest degradation rate could be determined. In general, the photodegradation efficiency will decrease with an increase in the pollutant initial concentration. A possible reason is that abundant pollutant molecules will adsorb on the catalyst surface when increasing the pollutant initial concentration. A large amount of the adsorbed pollutant molecules is thought to cause an adverse effect on the reaction of the pollutant molecules with the photoinduced active radicals, which could be attributed to the lack of direct contact between them. It should also be noted that the photons cannot reach the surface of the photocatalyst owing to the increase in pollutant concentration. These reasons could account for the decline in degradation efficiency. Furthermore, the light intensity enhancement enables the semiconductors to be motivated, which is beneficial for generat-

ing more active radicals and, thus, improving the degradation rate. However, the further increase in light intensity will have no obvious influence on the degradation performance of photocatalysts because the semiconductor has already been excited. Moreover, pH is one of the main influencing factors for the degradation of some organic pollutants. It can influence the surface state, interface potential, and surface charge of the photocatalyst. It is also a significant operational variable in the actual wastewater treatment [164]. Finally, the stability of the composite photocatalysts for a degradation system is crucial and indispensable. The ideal photocatalyst should be able to maintain a relatively high degradation performance after consecutive degradation tests. Therefore, it is very necessary to conduct recycling tests before putting the photocatalysts into practice. Typically, Zhang et al. [165] prepared novel $Fe_2O_3/TiO_2$/diatomite composites by using chemical precipitation–calcination and investigated their degradation performance towards ciprofloxacin (CIP). The effect of the operating parameters, including the CIP concentration, the catalyst dosage, the pH values, and the stability of the photocatalysts, were analyzed in detail, and the results are shown in Figure 13. Actually, there are many influencing factors on the performance of composite photocatalysts. It is hard to give a definite optimization sequence of the influencing parameters. However, it may be a feasible work to carry out orthogonal experiments in a specific experiment to determine the key preparation parameters.

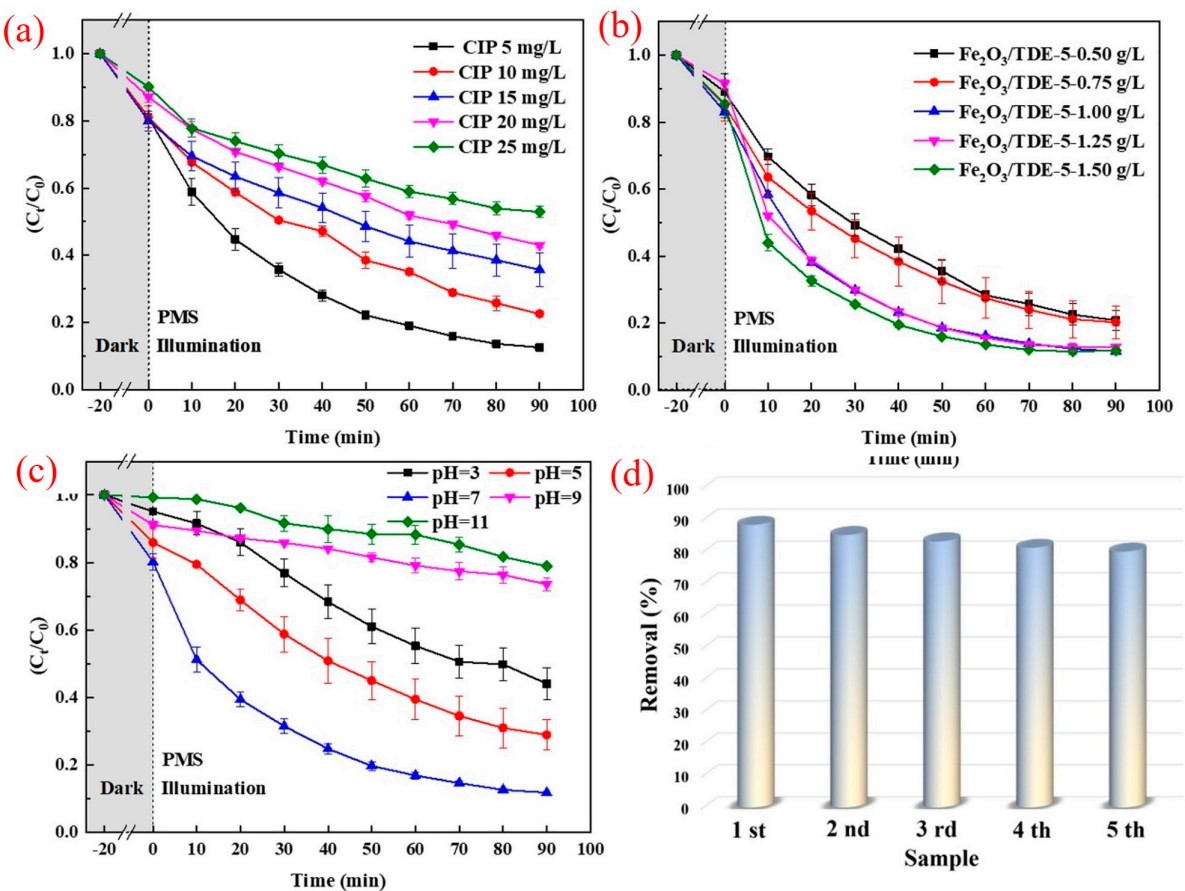

**Figure 13.** (**a**) Effect of CIP concentration. (**b**) Effect of PMS concentration. (**c**) Effect of catalyst dosage. (**d**) Effect of pH values on the removal of CIP and the reusability test results of $Fe_2O_3$/TDE-5 [165]. Copyright 2022 Elsevier.

## 8. Contaminant Degradation Mechanism

To investigate the degradation process of contaminants, the chemical oxygen demand (COD) is used to assess the quality of effluents by predicting their oxygen requirement [166,167]. To determine the intermediate products of the contaminant in the degradation process, LC-MS analysis is a useful method to analyze the degradation

products and path. Most of the intermediate compounds could be identified by the position of the peaks of m/z [168–170] (Figure 14). However, other undetected organic small molecules may exist in the solution owing to the insufficient separation performance of the chromatography column and low residual concentration. Moreover, HPLC can also be used to confirm the degradation extent of the pollutant [171].

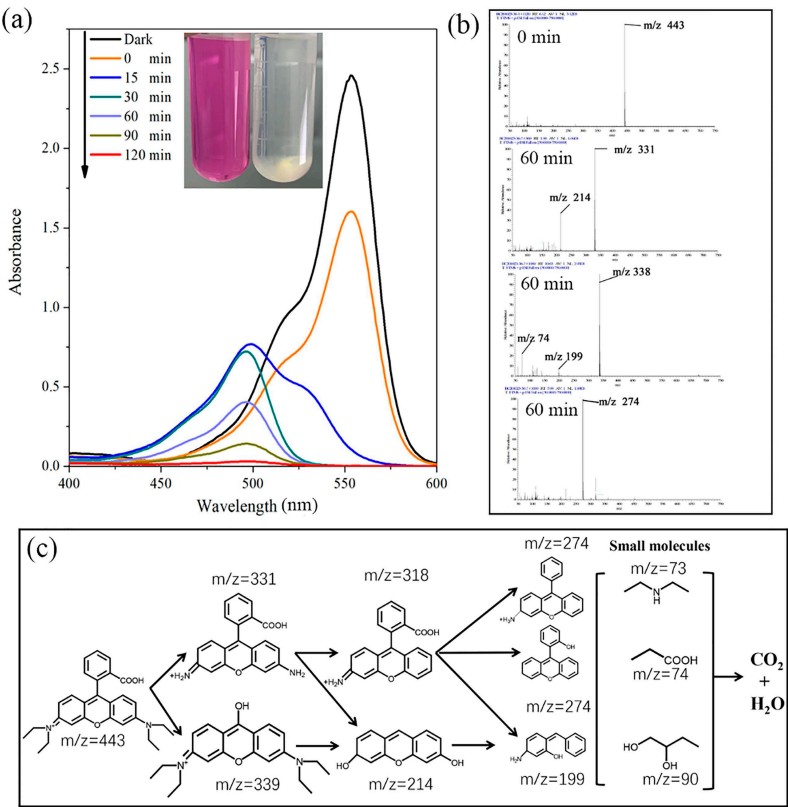

**Figure 14.** (**a**) UV–vis spectra of RhB under different illumination times, (**b**) representative mass spectra of RhB and degradation intermediates of RhB after 60 min of illumination, and (**c**) possible photocatalytic degradation routes of RhB under solar light [170]. Copyright 2021 Elsevier.

To study the photodegradation mechanism, the electron paramagnetic resonance (EPR) spectra were used to reveal the radical species involved in the photocatalytic process; 3,3,5,5 tetramethyl-1-pyrolline-N-oxide (TMPO) and 5,5 dimethyl-1-pyrolline-N-oxide (DMPO) were usually used as probes to detect the varieties of ROS such as hydroxyl radicals and superoxide radicals generated in the degradation reaction [172]. Moreover, active radical experiments could be carried out to test the effect of ROS in the photocatalytic system. For instance, silver nitrate ($AgNO_3$), edentate disodium (EDTA-2Na), isopropanol (IPA), and 1,4-benzoquinone (BQ) can be used to capture $e^-$, $h^+$, hydroxyl radicals, and superoxide radicals, respectively. The function of the ROS can be determined by observing the change in degradation efficiency after the addition of various scavengers [100]. The electron transfer process in the semiconductor/CP photocatalytic system can be summarized as follows: Under light irradiation, hydroxyl radicals could be produced owing to the diffusion of the photoinduced $h^+$ to the surface of the semiconductor/CP composite and the reaction of $h^+$ with the adsorbed water molecules. The hydroxyl radicals and $h^+$ could oxidize the pollutant molecules that are adsorbed on the surface of the composite photocatalysts. Meanwhile, electrons will react with the adsorbed $O_2$ to form superoxide radicals. The produced hydroxyl radicals and superoxide radicals with a strong oxidation ability could rapidly degrade the target pollutant molecules into small molecules that are non-toxic and harmless such as $CO_2$ and $H_2O$ (Figure 15).

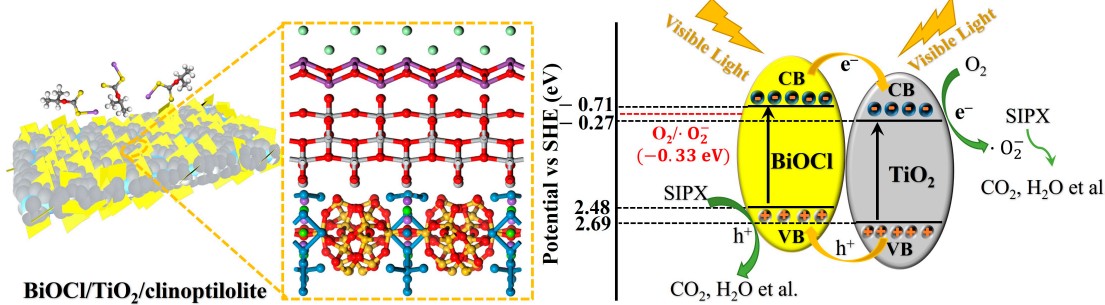

**BiOCl/TiO₂/clinoptilolite**

**Figure 15.** Possible mechanism for the photocatalytic degradation process of SIPX under visible light irradiation over the BTC nanocomposites [152]. Copyright 2021 Elsevier.

Furthermore, theoretical calculations are useful for revealing the microscopic reaction mechanism over the composite surface by using density function theory (DFT). The interface interaction and charge change between the pollutant and composite photocatalyst could be deeply investigated. Li et al. [173] unveiled the mechanisms of peroxymonosulfate (PMS) activation by a $NiFe_2O_4$/clinoptilolite composite using DFT simulations, and the hybrid catalyst exhibited a good photodegradation performance towards bisphenol A (BPA). A possible generation mechanism and transfer pathways of the radical over the $NiFe_2O_4$/clinoptilolite surface were illustrated in Figure 16 based on the results of the DFT analysis and EPR tests.

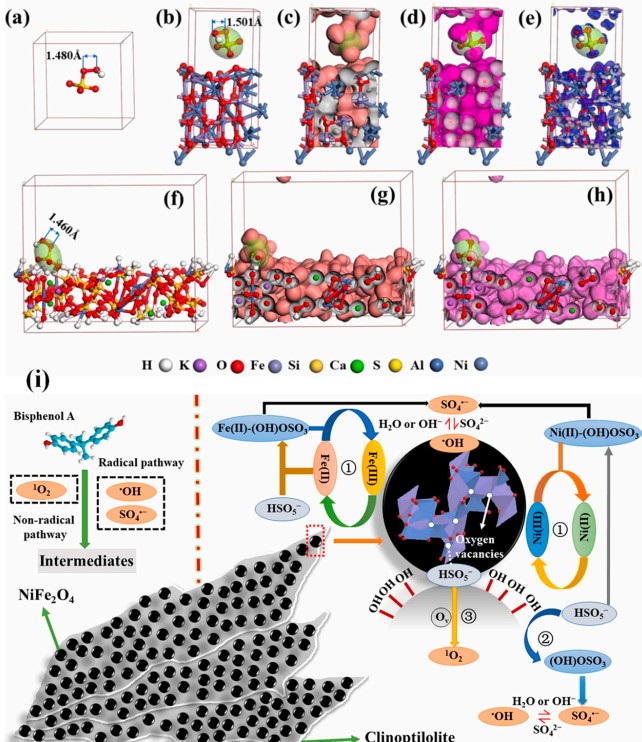

**Figure 16.** (**a**) The spatial stable state of PMS after geometry optimization; (**b–e**) the spatial stable state, charge density, STM profile, and charge density difference of PMS/$NiFe_2O_4$ after geometry optimization; (**f–h**) the spatial stable state, charge density, and STM profile of PMS/clinoptilolite after geometry optimization; and (**i**) the schematic diagram of the possible radical generation mechanism, as well as radicals' transfer pathways [173]. Copyright 2021 Elsevier.

## 9. Conclusions

We summarized the photodegradation efficiencies of different kinds of binary and ternary semiconductor/CP composites towards various EOPs in Tables S1 and S2. It can be

found that the deposition of photosensitive semiconductors on the surface of zeolite mineral supports showed a superior synergistic effect for the efficient removal of EOPs, which could overcome the shortcomings of bare semiconductor photocatalysts. Clinoptilolite is the most common and widely used mineral among the various natural zeolites. Generally, the synergistic effect of clinoptilolite supports and semiconductor photocatalysts could be explained by the following reasons: (1) clinoptilolite offers an enhanced adsorption ability to adsorb more pollutant molecules near the surface of catalysts and, thus, can improve the degradation reaction rate; (2) clinoptilolite could assist the dispersion of the bare semiconductor and, thus, avoid the serious agglomeration of crystalline grain and enhance the utilization rate of semiconductor-based catalysts; (3) clinoptilolite shows the obvious co-catalysis effect for accelerating the decomposition of the target pollutants; (4) abundant negative hydroxyl groups on the surface of clinoptilolite could effectively improve the separation of the photoinduced carriers by an electrostatic repulsion force; and (5) the transfer process of charges will be promoted and the recombination of $e^-/h^+$ pairs can be delayed due to the presence of Lewis acid (Al atoms)/base (oxygen sites) in the framework structure of clinoptilolite.

This review covers aspects of semiconductor-based composite photocatalysts by using clinoptilolite as the support in recent years and attempts to provide a summary regarding the effects of clinoptilolite support in semiconductor-based composite photocatalysts. The authors summarized the preparation methods and degradation of EOPs by clinoptilolite-based composite photocatalysts. The photodegradation efficiencies of clinoptilolite-based binary/ternary photocatalysts towards various kinds of EOPs were reported. Actually, almost all the EOPs have the characteristics of having a long persistence with long half-lives, potential bioaccumulation in fatty tissues, bio-magnification in food chains, and high toxicity with low concentrations. At the same time, current WWTPs cannot eliminate trace EOPs. Therefore, residual noxious contaminants will enter natural watershed water and soils, posing serious threats to humans and animals. The common synthesis and application parameters on the effects of the composites' photodegradation efficiencies, including the loading amount of semiconductor, the calcination temperature, the catalyst dosage, the pH, and the initial pollutant concentration, were discussed. Undoubtedly, both the synthesis parameters of composite photocatalysts and the application parameters in the photocatalysis process are closely related to the pollutant degradation efficiency. From the perspective of the structure–activity relationship between the composites' microstructures and their performance, the optimization of the synthesis parameters seems more meaningful, because the premise for optimizing the application parameters should be that the composite itself already has excellent photocatalytic potential. Furthermore, the authors introduced an approach to analyzing the possible decomposition products and degradation mechanisms of the target contaminants. It should be highlighted that all the intermediate products could finally be mineralized into non-toxic small molecular substances such as $CO_2$ and $H_2O$ over the zeolite mineral composites.

## 10. Outlook

Hybridizing clay with other semiconductors provides a new idea for preparing heterogeneous catalysts. Except for zeolite minerals, other kinds of clays such as kaolinite, montmorillonite, rectorite, and halloysite can also be used as catalyst supports, considering the advantages of having a low cost, strong adsorption ability, and chemical stability, and being environmentally friendly. It has been reported that clay minerals can effectively restrict the agglomeration of nano-scale catalysts and improve the dispersity of catalysts on their surface. The introduction of clay can regulate the morphologies and microstructures of the anchored semiconductors by interfacial effects. Furthermore, most of the semiconductor/clay composites have stable photocatalytic activity, mainly benefiting from a close interface combination between the semiconductor and clay surface via chemical bonding.

From the view of resource conservation, the regeneration and cyclic utilization of zeolite mineral composites is meaningful to sustainable development. To date, almost all of

the current clay-based composite photocatalysts are powder, leading to the hard recovery of the photocatalysts. The collection process may require high-speed centrifugation or prolonged natural sedimentation. The deposition of semiconductor photocatalysts over clay mineral-based porous ceramic substrates may be a good option for addressing this issue, and the ideal porous ceramic substrates can be arranged in a cluster to treat EOPs in WWTPs.

Though various methods have been proposed to synthesize the clinoptilolite-based photocatalysts, including sol–gel, hydrothermal, ion exchange, hydrolysis, chemical vapour deposition, impregnation, etc., it is hard and meaningless to evaluate which is the best route among all the synthesis approaches. Undoubtedly, all these methods have their own merits and demerits. Taking the precipitation method as an example, it is recognized as one of the desirable methods with which to prepare semiconductor NPs supported on clinoptilolite due to the feature of having a low cost and convenience. However, the size of the NPs is hard to control because of the high surface energy, and, thus, serious agglomeration will appear. That is to say, it is unwise to determine which preparation method is the best only by considering the cost and convenience while ignoring the photocatalytic performance. The degradation performance of the composite photocatalysts is the decisive factor that should be considered first. Given the industrial application, the expected zeolite mineral composites should degrade the target pollutants under solar light irradiation instead of using an artificial light source (UV or visible light). Clearly, developing new synthesis methods in order to achieve zeolite-based photocatalysts that would degrade pollutants under sunlight should be the mainstream research direction due to the economic and environmental factors. However, it remains challenging to achieve the large-scale application of composite photocatalysts because of some pending problems. Future work will still focus on preparing photocatalysts with a low cost, stable performance, and easy recycling, which are significant factors for practical application in WWTPs.

Considering the irreversible harm caused by toxic pollutants, all the current EOPs should be added to water quality standards according to their toxicological and epidemiological research. The long-term monitoring of EOP levels is necessary, which is a crucial reference for the water quality classification. Although most of the EOPs can be mineralized and decomposed into an inorganic small molecule ($CO_2$, $H_2O$) by zeolite mineral composites, the complete removal of EOPs is still challenging due to the complexity of pollutants and their derivative by-products. Moreover, it is predictable that more and more EOPs will be included in the scope of regulation with the development of more stringent water quality standards, which also presents new challenges for the current degradation technology.

**Supplementary Materials:** The following supporting information can be downloaded at: https: //www.mdpi.com/article/10.3390/catal14040216/s1; Table S1: Different kinds of binary semiconductor/CP composites and their photocatalytic efficiency in decomposing pollutants; Table S2: Different kinds of ternary semiconductor/CP composites and their photocatalytic efficiency in decomposing pollutants [174–176].

**Author Contributions:** Conceptualization, P.Z., Y.S. and J.L.; methodology, X.D.; software, F.W. and X.D.; validation, J.L., F.W. and P.Z.; formal analysis, Y.S.; investigation, X.C.; resources, Y.S. and J.L.; data curation, Y.S.; writing—original draft preparation, P.Z., S.Z. and X.C.; writing—review and editing, P.Z. and J.L.; visualization, X.D.; supervision, J.L.; project administration, J.L.; funding acquisition, J.L., F.W. and P.Z. All authors have read and agreed to the published version of the manuscript.

**Funding:** This research was funded by the National Key Research and Development Program of China (2023YFC3806102), the National Projects Funded by the Central Government to Guide Local Scientific and Technological Development of China (236Z4108G), and the Postdoctoral Funding Project of Hebei Province (B2023003019).

**Data Availability Statement:** The data reported were taken from publications included in the references.

**Acknowledgments:** The authors are thankful to Elsevier, MDPI, Springer, and American Chemical Society for copyright permission.

**Conflicts of Interest:** The authors declare no conflicts of interest.

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
