# Peer review of "Removal of Emerging Organic Pollutants by Zeolite Mineral (Clinoptilolite) Composite Photocatalysts in Drinking Water and Watershed Water"

_catalysts, doi:10.3390/catal14040216_

Round 1

Reviewer 1 Report

Comments and Suggestions for Authors

I have reviewed the manuscript entitled "Removal of emerging organic pollutants by zeolite mineral (clinoptilolite) composite photocatalysts: In drinking water and watershed water ", which presents a review based on zeolite photocatalyst for the removal of emerging organic pollutants.  I have some major suggestions to improve the quality of the manuscript, so the manuscript would be acceptable after addressing the following major issues:

Please rewrite the abstract and all sections of the article, using formal phrases. Utilize meaningful phrases to convey the study's objectives and findings accurately.

The abstract should be improved by adding the result based on the finding.

Please add some studies related to undoped titanium dioxide or zinc oxide with their limitations, can consider some studies to support your statements in introduction section such as Research on Chemical Intermediates 45, 2927-2945, Desalination and Water Treatment 161, 275-282, Applied Catalysis A: General 505, 507-514.

Please provide the heading and sub heading in more scientific ways.

References should be written according to the journal format

Please ensure the permission of each figure used in this review.

Please add more figures to enhance the interest of readers.

Please do the formatting correctly specially in case of sub and super script species like oxygen radicals etc.

Need to mention the full form before writing a short form of any term.

Avoid to mention the short form in heading or sub heading.

Need to mention about the zeolite, why author choose?

Need to add few line in each section regarding the critical prospects, rather than writing a sum up of previously published articles.

Need to revise some figures with high quality as they are presented in blurry form.

Need to check the grammar of article thoroughly.

Comments on the Quality of English Language

Need to check the grammar of article thoroughly.

Reviewer 2 Report

Comments and Suggestions for Authors

The authors have combined some information in the article, but cannot be said to have worked through this information as implied in this review. If we talk about the mineralogical and geological part, it is not structured at all, meager and poorly written. I didn’t like the absence of their own graphics and tables in the review, the almost complete absence of the structural (crystal-chemical) part. In my opinion, this text is not suitable for review.
The list of cited literayure is also questionable.
Please see other comments in attached file.

Reviewer 3 Report

Comments and Suggestions for Authors

In this study, the authors reviewed the literature on clinoptilolite composite photocatalysts for removing emerging organic pollutants from water. The work is interesting; however, the following issues need to be addressed prior to a possible publication:

1. Abstract:

I suggest the authors to replace the word “quickly”.

2. Introduction:

a) Lines 37-41 & 87-90: Do not use compound references. Change the text and citations to read in a similar way like in lines 42-44.

b) Lines 90-94 & 96-99: Provide literature.

3. Zeolite minerals

Lines 147-158: Provide literature.

4. Common semiconductor/clinoptilolite photocatalysts:

The information provided should be summarized in a Table (like Table 1). Provide lamp wavelength and power, pollutant concentration, and catalyst amount as well.

5. Preparation methods of zeolite mineral composite photocatalysts:

Table 1: Add information on pollutant concentration as well as lamp wavelength and power.

6. Antibiotics:

The excessive use of antibiotics has been associated with the rapid emergence of resistant strains; authors should comment on this providing relevant literature.

7. Conclusions:

The authors mentioned that “The common synthesis and application parameters on the effects of composites’ photodegradation efficiencies, including loading amount of semiconductor, calcination temperature, the catalyst dosage, pH and initial pollutant concentration, were discussed.”, however it is not clear which parameters are critical for the preparation of clinoptilolite-based photocatalysts and for the efficiency of photocatalytic systems. Finally, it is not clear which direction in zeolite-based photocatalysts synthesis is the most promising in photocatalysis.

Comments on the Quality of English Language

Minor editing of English language required.

Round 2

Reviewer 2 Report

Comments and Suggestions for Authors

I still don’t have the feeling that the material has been significantly developed; rather, it is still a compilation of data from articles and is more focused on the work of Chinese scientists (sorry, I have no preconceptions, but judging by the list of references). I was somewhat surprised by the answer “Chinese natural zeolite”; this is the first time I’ve heard minerals classified by country. I think that Chinese zeolite should not differ from zeolites in other countries. Although minerals grow in layers (always), zeolite is characterized by faceted crystals and a non-layered structure and texture (in general). I would advise the authors to get rid of this inaccuracy.

The crystal chemical part has been added as suggested (much better than in the original version).

I have no specific comments on the text, but there is a feeling that this text seems to have been collected (compiled) from other works without critical reasoning, comprehension and discussion. However, I believe that such a summarized material could be of interest to some readers.

Reviewer 3 Report

Comments and Suggestions for Authors

The authors addressed all issues raised.